Semaphorin4A promotes lung cancer by activation of NF-κB pathway mediated by PlexinB1

Wei Xiang 1
Liu Zhili 2
Shen Yili 1
Dong Hui 1
Chen Kai 1
Shi Xuefei 1
Chen Yi 1
Wang Bin 13757295077@139.com 1
Dong Shunli dongshunli@hzhospital.com 1
1 Department of Respiratory Medicine, Huzhou Central Hospital, Affiliated Central Hospital, Huzhou University , Huzhou , Zhejiang , China
2 Department of Oncology, The Jiangyin Clinical College of Xuzhou Medical University , Wuxi , Jiangsu , China
Qin Jiangjiang
Electronic publication date: 2023 Oct 24
Publication date: 2023
Volume: 11
Electronic Location ID: e16292
Received 2023 Aug 4; Accepted 2023 Sep 22
Copyright: ©2023 Wei et al.
Copyright year: 2023
Copyright holder: Wei et al.
License: This is an open access article distributed under the terms of the Creative Commons Attribution License, which permits unrestricted use, distribution, reproduction and adaptation in any medium and for any purpose provided that it is properly attributed. For attribution, the original author(s), title, publication source (PeerJ) and either DOI or URL of the article must be cited.
License URL: https://creativecommons.org/licenses/by/4.0/

Keywords: Semaphorin4A, Lung cancer, NF-κB pathway, PlexinB1, IL-6

Funding: Social Development Project of Public Welfare Technology Application in Zhejiang Province LGF21H160003 Public Technology Applied Research Program of Huzhou City Key Program 2019GZ35 This study was supported by the Social Development Project of Public Welfare Technology Application in Zhejiang Province (LGF21H160003) and the Public Technology Applied Research Program of Huzhou City (Key Program no. 2019GZ35). The funders had no role in study design, data collection and analysis, decision to publish, or preparation of the manuscript.

==============================
Background

Lung cancer (LC) is the most prevalent cancer with a poor prognosis. Semaphorin4A (Sema4A) is important in many physiological and pathological processes. This study aimed to explore the role and mechanism of Sema4A in LC.

Methods

Firstly, Sema4A expression was analyzed by the available dataset and detected in human normal bronchial epithelial cell line (HBE) and LC cell line (NCI-H460). Then, LC cells were transfected with Sema4A siRNA, and the cells were stimulated by PlexinB1, PlexinB2, PlexinD1 blocking antibodies, IgG antibody, BAY 11-7082 (an inhibitor for NF-κB pathway) and Sema4A-Fc protein, alone or in combination. After transfection, PlexinB1 mRNA expression was analyzed. Next, the biological functions, including proliferative, migratory, invasive abilities and viability of the cells were detected by colony formation, scratch, Transwell and MTT assays, respectively. NF-κB, Stat3 and MAPK protein expressions were determined by western blot. Furthermore, the secretion of IL-6 in LC cells was tested by ELISA.

Results

Sema4A was highly expressed in LC tissues and cells, could activate the NF-κB pathway and upregulate PlexinB1 mRNA expression. Furthermore, we observed that Sema4A knockdown suppressed the biological functions of NCI-H460 cells, while Sema4A-Fc protein reversed the situation. However, Sema4A-induced biological functions and activation in the NF-κB pathway were inhibited by PlexinB1 blocking antibody. Consistently, Sema4A promoted IL-6 production, which was down-regulated by PlexinB1 blocking antibody and BAY 11-7082.

Conclusions

Sema4A may facilitate LC development via the activation of the NF-κB pathway mediated by PlexinB1, suggesting that Sema4A would be a novel therapeutic target for LC treatment.

Introduction

Lung cancer (LC) is the leading cause of cancer-related death worldwide, particularly in some Asian countries, such as China (Hu et al., 2018) and Japan (Shirai et al., 2017). In the past few decades, despite some progress in LC therapy having been made, the prognosis of patients with LC remains poor (Takayuki et al., 2018). The overall five-year survival rate for LC patients is less than 20%, this is mainly due to the fact that the initial symptoms are not apparent and most LC patients are already at an advanced stage when diagnosed (Takayuki et al., 2018). The initiation and development of LC are associated with multiple intracellular events, such as the activation of various oncogenes and the inactivation of some tumor-suppressing genes (Min et al., 2018). Thus, tumor endogenous factors, which seem to result in the development of LC, have gained overwhelming attention. Identification of these endogenous factors can not only better understand the initiation and progression of LC, but also provide novel targets for the treatment of LC.

Semaphorins are a big family with over 30 members, and all of them contain a highly conserved N-terminal domain called as Sema domain (Rezaeepoor et al., 2018). Plexins have been determined as the best-characterized receptors for semaphorins, which are grouped into four sub-families consisting of nine members (Wylie et al., 2017). It has been unraveled that some semaphorins can bind directly with plexins and activate Plexin-regulated signal transduction (Singh et al., 2019). Although semaphorins as well as plexins were initially described as components of the regulatory system responsible for guiding axons during central nervous system development, accumulating evidence indicates that some semaphorins, by interacting with their receptors, exert a regulatory effect in the initiation and progression of tumor (Singh et al., 2019). Semaphorin4A (Sema4A) has been found to bind to different receptors (including PlexinB1, PlexinB2, PlexinD1), and PlexinB1, PlexinB2, PlexinD1 can affect various pathways that are related to cellular invasion, migration and growth in cancers (Carvalheiro et al., 2020). Peacock et al. (2018) have reported that Sema3C can facilitate the growth of cancer cells while inhibiting Sema3C can delay the development of castration-resistant prostate cancer (CRPC) by regulating PlexinB1. Meanwhile, a study has reported that Sema4D/PlexinB1 may block tumor blood supply by vasculogenic mimicry pattern, which halts the development of NSCLC (non-small cell lung cancer) eventually (Xia et al., 2019). In addition, activating the NF-κB pathway can promote the invasion and metastasis of LC cells (Wang et al., 2021). Blocking the NF-κB pathway can suppress LC cell growth in vivo (Xu et al., 2016). The NF-κB pathway is thus a pivotal regulator for the development and progression of LC. Previous studies have reported that Sema4A upregulation is associated with the NF-κB pathway in osteoarthritis (Zhang et al., 2019) and rheumatoid arthritis (Wang et al., 2015). PlexinB1 has the ability to activate MAPK, which is the upstream component of the NF-κB pathway (Aurandt, Li & Guan, 2006). Additionally, through binding to PlexinB1 via AKT/NF-KB cascade, Sema 4D can induce C-X-C motif chemokine ligand 9/C-X-C motif chemokine ligand 10 production in vitro (Ke et al., 2017). However, the exact relationship among Sema4A, PlexinB1 as well as NF-κB pathway in LC is still unclear.

Hence, in this study, to explore whether Sema4A promoted LC development by the PlexinB1-mediated NF-κB pathway, we treated LC cells with various stimulations, and then focused on the biological activity and the expression of the NF-κB pathway, thereby providing a novel target for LC treatment.

Materials and Methods

Bioinformatics analysis

The Sema4A mRNA expression in LC was obtained from the publicly available dataset (The Cancer Genome Atlas project, TCGA, https://ualcan.path.uab.edu/cgi-bin/TCGAExResultNew2.pl?genenam=SEMA4A&ctype=LUSC), which included 52 normal lung tissues and 503 LC tissues.

Cell culture

Human bronchial epithelioid (HBE) cells and LC (NCI-H460) cells were obtained from uhan Pu-nuo-sai Life Technology Co. Ltd. (CL-0346; CL-0299; Wuhan, China). Based on the instruction, HBE and NCI-H460 cells were cultured in Dulbecco’s modified Eagle medium (DMEM) supplemented with 10% fetal bovine serum (FBS) in a 37 °C incubator with 5% CO2.

si-RNA transfection in NCI-H460 cells

The small interfering RNA (siRNA) transfection was conducted as previously described (Zhong et al., 2013). Briefly, NCI-H460 cells were seeded into 12-well plates before the day of transfection. The next day, following the manufacturer’s instruction, NCI-H460 cells were transfected with Sema4A-specific siRNA (si-Sema4A) or control siRNA (si-NC) using Lipofectamine RNAiMAX Reagent. After 48 h of transfection, the cells were collected for subsequent experiments.

Stimulation assays

The untransfected NCI-H460 cells seeded in 12-well plates were stimulated by the following agents: PlexinB1 (MAB37491), PlexinB2 (AF5329), PlexinD1 blocking antibodies (AF4160), IgG antibody (1-001-A), 10 uM BAY 11-7082 (an inhibitor for NF-κB pathway, HY-13453; MedChemExpress, Monmouth Junction, NJ, USA) and Sema4A-Fc protein (0–100 nmol/L, KMPH3353; KMD Bioscience, Tianjin, China), alone or in combination. The untransfected cells without any stimulation were utilized as the control group. In addition, the transfected NCI-H460 cells were treated with or without 100 nmol/L Sema4A-Fc protein. The PlexinB1, PlexinB2, PlexinD1 blocking antibodies and IgG antibody were supplied by R&D Systems (Minneapolis, MN, USA).

qPCR

The expressions of Sema4A and PlexinB1 mRNA were analyzed using quantitative polymerase chain reaction (qPCR). In short, the total mRNAs of the cells were extracted using a Trizol reagent (B511311, Sangon Biotech, Shanghai, China). Then, a NanoDrop ND-1000 spectrophotometer (NanoDrop, Wilmington, DE, USA) was applied to determine the quantity and purity of the mRNA. Afterwards, cDNA was synthesized with an RNA reverse-transcription kit (CW2569, CWBIO, Beijing, China) based on the manufacturer’s protocols. Subsequently, SYBR Premix Ex TaqII (RR820A; Takara, Dalian, China) was applied to conduct qPCR in the PCR instrument (Mastercycler; Eppendorf, Hamburg, Germany). The amplification conditions were as follows: predenaturation at 95 °C for 10 min, 40 cycles of denaturation at 95 °C for 15 s and annealing/extension at 60 °C for 1 min. The relative Sema4A and PlexinB1 mRNA expressions were measured, and GAPDH was employed as a reference gene. The sequences of qPCR primers utilized in this study are presented in Table 1.

Table 1 qPCR primers.

Gene	Forward primer	Reverse primer	
Human Sema4A	TGGATGGGATGCTCTATTCTGG	GCGGAGGAAGTTGTCGGTC	
Human PlexinB1	TCCACCAACTGCATTCACTC	GTGACCTTGTTTTCCACAGCAG	
Human GAPDH	TGTGGGCATCAATGGATTTGG	ACACCATGTATTCCGGGTCAAT	

Western blot analysis

The total proteins of the cells were prepared by radioimmunoprecipitation assay (RIPA) buffer. A bicinchoninic acid (BCA) assay was then used to determine the protein concentrations. After that, equal amounts of protein samples were subjected to 5% sodium dodecyl sulfate-polyacrylamide gel electrophoresis (SDS-PAGE) and transferred onto polyvinylidene fluoride (PVDF) membranes. Upon blocking with 5% non-fat milk, the membranes were probed overnight with primary antibodies against Sema4A (1:1,000, ab70178), p-NF-κB (1:1,000, AF2006), NF-κB (1:1,000, AF5006), p-Stat3 (1:1,000, AF3293), Stat3 (1:1,000, AF6294), p-MAPK (1:2,000, DF7632), MAPK (1:2,000, DF8796), and GADPH (1:5,000, ab8245) at 4 °C. Following washing, the membranes were incubated with HRP-conjugated secondary antibodies for another 1.5 h. Finally, the protein signals were visualized with electrochemiluminescence (ECL) reagents and quantified by ImageJ software. All the primary antibodies were provided by Affinity, except for GADPH and Sema4A from Abcam.

Colony formation assay

To assess cell proliferation ability, clonogenic assays were conducted. First of all, cells were seeded in 12-well plates at a density of 500 cells per well. After performing corresponding treatments and culturing for 2 weeks to form colonies, the cells were washed with phosphate-buffered saline (PBS) and fixed with 4% paraformaldehyde before staining with 0.1% crystal violet. Finally, the number of colonies containing more than 50 cells was calculated using a camera.

Cell migration test

The cellular migratory capacity was evaluated via a scratch assay. In brief, a marker pen was used to draw horizontal lines on the back of 6-well plates with an interval of about 0.5–1 cm. Following that, cells were seeded in 6-well plates and cultured until confluent. Subsequently, scratches were created on cells in the horizontal lines perpendicular to the back with a pipette tip. Hereafter, cellular fragments were removed by PBS carefully, and the remaining cells were treated with different stimulations and incubated in serum-free mediums for another 24 h. Finally, the wound areas were captured with a microscope and assessed by ImageJ. The relative migration distance was calculated based on the equation: (scratch width of 0 h-scratch width of 24 h)/scratch width of 0 h ×100%.

Cell invasion assay

The invasion ability of the cell was assessed by Transwell assay. To begin with, 30 µL of diluted Matrigel was used to precoate the upper Transwell chambers at 4 °C overnight. Hereafter, the cells treated with different treatments were added to the upper chambers. DMEM medium containing 10% FBS was injected into the lower chambers. After incubation for 24 h, the non-invading cells on the surface of the upper chamber were swabbed with cotton buds. The invading cells on the surface of the lower chamber were fixed with formaldehyde (4%), stained with crystal violet (0.1%), and counted with a microscope.

MTT assay

For measuring cell viability, 3-(4,5-dimethylthiazol-2-yl)-2,5-diphenyltetrazolium bromide (MTT) assays were performed based on the manufacturer’s protocol. Briefly, cells were plated evenly in 96-well plates at a density of 5,000 cells/well for 48 h to allow cells to adhesion. Subsequently, the cells were administrated with different treatments and exposed to 10 µL of MTT solution (E606334-0500; BBI Life Sciences, Shanghai, China). After a 2 h incubation, DMSO was added to each well to dissolve the produced formazan crystal. The absorbance value of each well was detected at 490 nm by a microplate reader (CMaxPlus; Molecular Devices, San Jose, CA, USA).

ELISA

Enzyme-linked immunosorbent assay (ELISA) was applied to quantify the level of IL-6 in the cell supernatant. Following various treatments, the cell culture medium was harvested and centrifuged to remove cellular debris. Then, based on the manufacturer’s protocols, the supernatant was collected for the detection of IL-6 level by ELISA kit (MM-0049H2; Jiangsu Meimian Industrial Co., Ltd., Jiangsu, China). The absorbance value was detected at 450 nm.

Statistical analysis

The data of the study were presented as mean ± SD, and analyzed by SPSS 19.0. One-way analysis of variance (ANOVA) and Tukey tests were applied for multi-group comparison, and two-tailed Student’s-test was utilized for intergroup comparisons. Kruskal–Wallis H test was applied, if variances were not equal. Statistical significance was set at p < 0.05.

Results

Sema4A was highly expressed in LC

By analyzing the data of 52 normal lung tissues and 503 LC tissues from TCGA, we observed that Sema4A expression was obviously increased in the LC tissues (P < 0.01, Fig. 1A). Then, we compared the expression level of Sema4A mRNA and protein in human bronchial epithelioid (HBE) cells and LC (NCI-H460) cells. The results of qPCR revealed that NCI-H460 cells markedly exhibited a higher level of Sema4A mRNA than HBE cells (P < 0.01, Fig. 1B). In line with this, on western blot analysis, the expression of Sema4A protein in NCI-H460 cells was also higher than that in HBE cells (P < 0.01, Fig. 1C).

Figure 1 Sema4A expression was increased in LC.

(A) Sema4A expression was analyzed by TCGA dataset. (B) Sema4A mRNA expression was tested by qPCR. (C) Sema4A protein expression was measured by western blot. ∧P <0.05, and ∧∧P < 0.01 vs. Normal; &P < 0.05, and &&P < 0.01 vs. HBE cells. Results were presented as mean ± SD. n = 3. Note: Sema4A: semaphorin 4A; LC, lung cancer; TCGA, The Cancer Genome Atlas project; qPCR, quantitative polymerase chain reaction; HBE: human bronchial epithelioid.

Sema4A upregulated PlexinB1 mRNA expression, and promoted the growth and motility of LC cells

Then, we investigated whether Sema4A could function as an oncogene in LC. We treated LC cells with varying concentrations of Sema4A-Fc protein (ranging from 10–100 nmol/L). From the western blot, we observed that the phosphorylation of NF-κB pathway-related proteins (including NF-kB, Stat3 and MAPK) increased as the increased Sema4A-Fc protein concentration, and the obvious effect was observed at a concentration of 100 nmol/L (p < 0.05, Fig. 2). Thus, 100 nmol/L of Sema4A-Fc was selected as the final concentration for all subsequent experiments.

Figure 2 Sema 4A activated the NF-κB pathway in LC cells.

Upon treatment with various concentrations of Sema4A-Fc protein, the phosphorylation of NF-κB, Stat3 and MAPK in LC cells were determined by western blot. @P <0.05, and @@P < 0.01 vs. 0 nmol/L. Results were presented as mean ± SD. n = 3. Note: Sema4A: semaphorin 4A; LC, lung cancer.

Next, we investigated whether the mRNA expression of PlexinB1 was regulated by Sema4A in LC cells. The results displayed in Fig. 3 found that after transfection with si-Sema4A, the expression of PlexinB1 mRNA was downregulated (p < 0.05). However, treatment with 100 nmol/L of Sema4A-Fc protein led to a significant upregulation of PlexinB1 mRNA expression (p < 0.05).

Figure 3 Sema4A upregulated PlexinB1 mRNA expression for LC cells.

After transfected with si-NC or si-Sema4A, the PlexinB1 mRNA expression of LC cells treated with or without Sema4A-Fc protein were tested by qPCR. ▴P < 0.05, and ▴▴P < 0.01 vs. si-NC; + P < 0.05, and ++P < 0.01 vs. si-Sema4A. Results were presented as mean ± SD. n = 3. Note: Sema4A: semaphorin 4A; LC, lung cancer.

We examined the biological activities of human LC cells. As presented in Fig. 4, the in vitro proliferative, migratory and invasive abilities as well as viability were obviously reduced after NCI-H460 cells were transfected with si-Sema4A (p < 0.01). To further explore whether Sema4A could modulate the biological activities of LC cells, NCI-H460 cells transfected with si-Sema4A were treated with 100 nmol/L of Sema4A-Fc protein. The results showed that si-Sema4A-mediated downregulation of biological activities in NCI-H460 cells was evidently rescued by Sema4A-Fc protein (p < 0.01).

Figure 4 Sema4A promoted the growth and motility of LC cells.

After transfected with si-NC or si-Sema4A, the proliferative, migratory, invasive abilities and the viability of LC cells treated with or without Sema4A-Fc protein were detected by colony formation (A), scratch (B), Transwell (C) and MTT (D) assays, respectively. Magnification for (B): ×40, magnification for (C): ×200. ▴P < 0.05, and ▴▴P < 0.01 vs. si-NC; +P < 0.05, and ++P <0.01 vs. si-Sema4A. Results were presented as mean ± SD. n = 3. Note: Sema4A: semaphorin 4A; LC, lung cancer; MTT: 3-(4,5-dimethylthiazol-2-yl)-2,5-diphenyltetrazolium bromide.

Sema4A promoted the development of LC in vitro by up-regulating PlexinB1

To analyze the effect of PlexinB1, PlexinB2 and PlexinD1 in Sema4A-induced biological activities, NCI-H460 cells were stimulated with 100 nmol/L Sema4A-Fc protein, PlexinB1 blocking antibody, PlexinB2 blocking antibody, PlexinD1 blocking antibody and IgG antibody alone or in combination. The results of Figs. 5 and 6 demonstrated that Sema4A-Fc protein remarkably up-regulated the proliferation, migration, invasion, and viability of LC cells (p < 0.05). However, the situation was completely reversed upon treatment with PlexinB1 blocking antibody (p < 0.05), while PlexinD1 blocking antibody only inhibited Sema4A-induced cellular migration (p < 0.05).

Figure 5 Sema4A facilitated the proliferative and migratory abilities of LC cells by upregulating PlexinB1.

After stimulation by Sema4A-Fc protein, PlexinB1, PlexinB2, PlexinD1 blocking antibodies, and IgG antibody alone or in combination, the proliferative and migratory abilities of LC cells were assessed by colony formation (A) and scratch (B) assays, respectively. Magnification for (B): ×40. @P < 0.05, and @@P <0.01 vs. Control; #P < 0.05, and ##P < 0.01 vs. IgG; −P < 0.05, and −−P < 0.01 vs. Sema4A-Fc+IgG. Results were presented as mean ± SD. n = 3. Note: Sema4A: semaphorin 4A; LC, lung cancer.

Figure 6 Sema4A facilitated the invasive ability and the viability of LC cells by upregulating PlexinB1.

After stimulation by Sema4A-Fc protein, PlexinB1, PlexinB2, PlexinD1 blocking antibodies, and IgG antibody, alone or in combination, the invasive ability and the viability of LC cells were assessed by Transwell (A) and MTT (B) assays, respectively. Magnification for (B): ×200. @P <0.05, and @@P < 0.01 vs. Control; #P < 0.05, and ##P <0.01 vs. IgG; −P < 0.05, and −−P < 0.01 vs. Sema4A-Fc+IgG. Results were presented as mean ± SD. n = 3. Note: Sema4A: semaphorin 4A; LC, lung cancer; MTT: 3-(4,5-dimethylthiazol-2-yl)-2,5-diphenyltetrazolium bromide.

Sema4A promoted NF-κB, Stat3 and MAPK phosphorylation via enhancing PlexinB1

To clarify the molecular mechanism of Sema4A in the development of LC, we stimulated NCI-H460 cells with PlexinB1 blocking antibody, 100 nmol/L of Sema4A-Fc protein, alone or in combination, and then detected the expressions of proteins associated with the NF-κB pathway. The results presented in Fig. 7 found that in comparison to the controls, the cells treated with 100 nmol/L of Sema4A-Fc protein had higher phosphorylation of NF-κB, Stat3 and MAPK (p < 0.01), which was similar to the results depicted in Fig. 2. However, PlexinB1 blocking antibody could reverse the situation (p < 0.05).

Figure 7 Sema4A promoted NF-κB, Stat3 and MAPK phosphorylation via upregulating PlexinB1 in LC cells.

After stimulating with PlexinB1 blocking antibody, Sema4A-Fc protein, alone or in combination, the phosphorylation of NF- κ B, Stat3 and MAPK in LC cells were evaluated by western blot. @P < 0.05, and @@P < 0.01 vs. Contol; ∗P < 0.05, and ∗∗P < 0.01 vs. Sema4A-Fc. Results were presented as mean ± SD. n = 3. Note: Sema4A: semaphorin 4A; LC, lung cancer.

Sema4A elevated IL-6 production by up-regulating the NF-κB pathway in LC cells

The production of the IL-6 was tested in Sema4A-Fc protein-treated LC cells stimulated with IgG antibody, PlexinB1 blocking antibody or BAY 11-7082 (an inhibitor for the NF-κB pathway). Interestingly, treatment with Sema4A-Fc protein significantly increased the secretion of IL-6 in NCI-H460 cells (p < 0.01, Fig. 8). As expected, after stimulation with PlexinB1 blocking antibody or BAY 11-7082, there was an obvious reduction in IL-6 production in NCI-H460 cells (p < 0.01).

Figure 8 Sema4A increased IL-6 production by upregulating the NF-κB pathway in LC cells.

After stimulation with Sema4A-Fc protein, the LC cells were treated with PlexinB1 blocking antibody, 10 uM BAY 11-7082 (a NF- κ B pathway inhibitor) or IgG antibody, the content of IL-6 in LC cells was estimated by ELISA. @P < 0.05, and @@P < 0.01 vs. Contol; ∗P < 0.05, and ∗∗P < 0.01 vs. Sema4A-Fc+IgG. Results were presented as mean ± SD. n = 3. Note: Sema4A: semaphorin 4A; LC, lung cancer; ELISA, Enzyme-linked immunosorbent assay.

Discussion

Semaphorin is a big family comprising more than 30 members, the member of the family is characterized by a structure called as Sema domain (Xu et al., 2021). In the past few years, growing evidence has indicated that some semaphorin family members are pivotal in the regulation of cell functions related to cancer (Suga et al., 2021). Additionally, it is reported that Sema4A not only plays a vital role in the activation and differentiation of T cells, but also is crucial in the regulation of the Th1/Th2 immune response (Kayama et al., 2019). More importantly, a published experiment has suggested that knockdown of Sema4A can improve the EMT process and increase sensitivity to doxorubicin (a chemotherapy agent for treating hepatoma) in hepatoma cells (Pan, Wang & Ye, 2016). Consistently, in the present study, we found that Sema4A was highly expressed in human LC cells, and regulating Sema4A expression could modulate the biological activity of human LC cells, which implied that Sema4A may be employed as a critical target for LC treatment.

Plexins are initially featured by their role as the receptors for semaphorins in the wiring of the neural network (Fazzari et al., 2007). Yet, mounting findings later have prompted researchers to re-evaluate and modify the biological functions of Plexins. Firstly, it has been discovered that numerous Plexins are widely expressed outside the nervous system, such as in tumor tissues (Kandemir et al., 2020). Additionally, with the revelation of signal transduction mechanisms, it is clear that Plexins can affect various pathways that are related to cellular invasion, migration and growth (Servage et al., 2020). Research conducted by Guan and co-workers has demonstrated that PlexinD1 is highly expressed in the liver tissues of patients with hepatocellular carcinoma, furthermore, PlexinD1 expression is correlated with many clinical characteristics, such as tumor hemorrhage and tumor grade (Li et al., 2020). In addition, PlexinB1 has been identified as a target gene directly correlated with the progression of prostate cancer in vitro (Liu et al., 2017). Most importantly, some studies have revealed that the functions of Plexins in cancers are regulated by semaphorins, including Sema3C (Peacock et al., 2018), Sema4D (Lontos et al., 2018), and Sema5A (Saxena et al., 2018). In this study, we found that regulating Sema4A expression could modulate PlexinB1 mRNA expression. In addition, we also observed that after blocking PlexinB1, Sema4A-induced proliferative, migratory and invasive abilities as well as viability of LC cells were suppressed evidently, but blocking PlexinB2 and PlexinD1 did not always significantly inhibit Sema4A-induced proliferative, migratory and invasive abilities as well as viability for LC cells, these observations implicated that Sema4A might contribute to the development of LC by interacting with PlexinB1. However, it should be noted that the impact of PlexinB1 in cancer is highly context-dependent, and in some cases, PlexinB1 can also inhibit the development of tumors (Jiang et al., 2021).

To further elucidate the underlying mechanism of Sema4A in LC, we examined intracellular signal transduction pathways. A report has revealed that the NF-κB pathway is aberrantly activated in various human cancers, such as lymphoma, liver, breast, colon and pancreatic cancer (Gaptulbarova et al., 2020). A putative response of NF-κB signaling activation can induce the production of cytokines, which contributes to the recruitment of immune cells to the tumor sites, thereby promoting the development of the tumor (Guo et al., 2020). The activated NF-κB pathway is strongly associated with tumor recurrence, drug resistance and a poor prognosis (Chen et al., 2020). Moreover, a published study has uncovered that Sema4D elicits oral lichen planus CD8+ T-cell migration by binding to PlexinB1 via AKT/NF-KB cascade (Ke et al., 2017). Stat3 exhibits pro-tumor effects in many cancers, whose target genes include NF-κB (Zhou & Chen, 2021). Here, in this study, we found that stimulating LC cells with Sema4A-Fc protein could activate the phosphorylation of NF-κB, Stat3 and MAPK, however, the situation was reversed upon treatment with PlexinB1 blocking antibody, which suggested that the specific mechanism of Sema4A facilitates LC development may link to PlexinB1 controlled the Stat3/NF-κB pathway.

IL-6 is a kind of important cytokine that facilitates the migration and proliferation of LC cells (Ritzmann et al., 2019). After being stimulated by inflammatory mechanisms or tumor microenvironment, mesenchymal stem cells (MSCs) can secrete IL-6, initiating epithelial-mesenchymal transition (EMT), thereby further promoting the metastasis of LC cells (Wang et al., 2017). Research has reported that the level of IL-6 is high in the serum of patients with breast cancer and LC, and the high IL-6 level is correlated with poor clinical prognosis (Yang et al., 2021). Furthermore, IL-6 production is NF-κB-dependent and is essential for the cytokine network to trigger and exacerbate LC development (Weil et al., 2019). A study has reported that Sema4A can aggravate the pathological progression of rheumatoid arthritis by synergistically regulating the generation of IL-6 with lipopolysaccharide (Wang et al., 2015). This study consistently presented that the level of IL-6 was upregulated by Sema4A-Fc protein treatment, nevertheless, upon stimulation with PlexinB1 blocking antibody or NF-κB pathway inhibitor, the level of IL-6 tended to the controls. The specific process of Sema4A acting on LC development and its possible mechanism was presented in Fig. 9, which further revealed that Sema4A may facilitate LC development via activation of NF-κB pathway mediated by PlexinB1.

Figure 9 The mechanism model diagram of Sema4A promotes LC development.

Note: Sema4A, semaphorin 4A; LC, lung cancer; the red upward arrow indicates an increase.

Finally, a potential limitation of the research should be mentioned. The absence of animal experiments is the major defect of the present study. Therefore, in the future, we will continue to validate the function and mechanism of Sema4A in LC with animal models.

Conclusions

Our results revealed that Sema4A exerted its oncogenic function in LC cells, and the mechanisms might contribute to the activation of the NF-κB pathway mediated by PlexinB1, whether the mechanism was related to PlexinB2 and PlexinD1 need further confirmed. This refined the current understanding regarding the mechanism involved in LC and suggested that Sema4A may be a potential target for treating LC.

Supplemental Information

Supplemental Information 1 The source data of the study

Click here for additional data file.

Supplemental Information 2 The original gels of western blot for the study

Click here for additional data file.

Additional Information and Declarations

Competing Interests

Author Contributions

Data Availability

The authors declare there are no competing interests.

Xiang Wei performed the experiments, prepared figures and/or tables, and approved the final draft.

Zhili Liu analyzed the data, prepared figures and/or tables, authored or reviewed drafts of the article, and approved the final draft.

Yili Shen performed the experiments, prepared figures and/or tables, and approved the final draft.

Hui Dong performed the experiments, prepared figures and/or tables, and approved the final draft.

Kai Chen analyzed the data, prepared figures and/or tables, and approved the final draft.

Xuefei Shi analyzed the data, prepared figures and/or tables, and approved the final draft.

Yi Chen analyzed the data, authored or reviewed drafts of the article, and approved the final draft.

Bin Wang conceived and designed the experiments, authored or reviewed drafts of the article, and approved the final draft.

Shunli Dong conceived and designed the experiments, authored or reviewed drafts of the article, and approved the final draft.

The following information was supplied regarding data availability:

The raw measurements are available in the Supplementary Files.

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
