# Peer review of "Semaphorin4A promotes lung cancer by activation of NF-κB pathway mediated by PlexinB1"

_PeerJ, doi:10.7717/peerj.16292_

## Round 0.1 · original submission · Major Revisions

Please carefully read the comments and suggestions from the reviewers and provide your point-to-point responses accordingly.

Reviewer 1 ·

Basic reporting

In this study, the authors tried to explore the relationship of between the deregulated Sema4A and NF-kB pathway in the development of lung cancer. It is an interesting work. Here are my comments.

Experimental design

The experimental design is good.

Validity of the findings

1. In introduction part, I only found the supporting evidence for connection between Sema4A and NF-kB signaling pathway, but not for between PlexinB1 and Sema4A. It is better to include introduction of the functional role of PlexinB1 in NF-kB activation.
2. Is there any clinical evidence supporting the deregulated expression of Sema4A in lung cancer? (Such as analysis results from TCGA data)
3. For the conclusion that “Sema4A promoted the development of LC in vitro by up-regulating PlexinB1”. Is there any direct evidence that the expression of PlexinB1 upregulated by Sema4A?
4. As both NF-kB and Stata3 were activated by Sema4A, please discuss the potential role of Sema4A- PlexinB1-stat3/NF-kB for LC development.

Reviewer 2 ·

Basic reporting

The present study investigated the role and mechanism of Sema4A in LC. The authors found that Sema4A may facilitate LC development via the activation of the NF-κB pathway mediated by PlexinB1. Although the subject is interesting, the manuscript needs some revision before further consideration for publication can be given. My concerns are listed as follows:
1. In Figure 1, the mRNA and protein overexpression multiples of Sema4A were not consistent.
2. Please provide the formula for calculating the relative migration distance in Cell migration test.
3. Since the authors believe that Sema4A can regulate PlexinB1, PlexinB1 expression should be tested among the si-NC, si-Sema4A, and si-Sema4A+Sema4A-Fc groups.
4. Figure 4-5 shown that after blocking PlexinB1, PlexinB2, or PlexinD1, Sema4A-induced proliferative, migratory and invasive abilities as well as viability of LC cells were suppressed. Although PlexinB1 has the most significant effect, the author should not focus only on PlexinB1 in the title and conclusion of the manuscript without explanation, while selectively ignoring PlexinB2 and PlexinD1.
5. Please check the results in Figure 2 and Figure 6, if the results exhibited the phosphorylation of NF-κB, Stat3 and MAPK, the ratio of NF-κB/GAPDH, Stat3/GAPDH and MAPK/GAPDH should be deleted.
6. In Fig. 7, the results of Sema4A-Fc+IgG should be ranked after Sema4A-Fc, while Sema4A-Fc+a PlexinB1 and Sema4A-Fc+BAY 11-7082 should be compared with Sema4A-Fc+IgG.
7. The manuscript needs to be passed through a carefully revised to eliminate grammatical errors and sentence corrections.
8. Why detect the PlexinB1 and PlexinD1 when the study is about the relationship between Sema4A and PlexinB2? This should be explained.
9. In the discussion section, it is recommended to delete the description of natural sources. The experimental results related to it are not reflected in the manuscript.

Experimental design

Good

Validity of the findings

Good

---

## Round 0.2 · Minor Revisions

The reviewers have provided some minor comments. Please provide your responses accordingly.

Reviewer 1 ·

Basic reporting

All my concerns has been addressed.

Experimental design

Good.

Validity of the findings

Good.

Additional comments

Please check if there is any statistical significance test missing.

Reviewer 2 ·

Basic reporting

The author basically answered my question satisfactorily, but there are two small corrections that need to be made:
1. Figure 1A does not show p-values;
2. page 11, line 29; page 12, line 6. Whether PlexinB2 and PlexinB1 are confused.

Experimental design

Good

Validity of the findings

Good

---

## Round 0.3 · accepted · Accept

The authors have addressed the concerns of the reviewers and the paper may be accepted at the current stage.